# A Combined Experimental and Computational Study on the Adsorption Sites of Zinc-Based MOFs for Efficient Ammonia Capture

**DOI:** 10.3390/molecules27175615

**Published:** 2022-08-31

**Authors:** Dongli Zhang, Yujun Shen, Jingtao Ding, Haibin Zhou, Yuehong Zhang, Qikun Feng, Xi Zhang, Kun Chen, Pengxiang Xu, Pengyue Zhang

**Affiliations:** 1Academy of Agricultural Planning and Engineering, Key Laboratory of Technologies and Models for Cyclic Utilization from Agricultural Resources, Ministry of Agriculture, Beijing 100125, China; 2School of Advanced Manufacturing, Guangdong University of Technology, Jieyang 515200, China; 3State Key Laboratory of Power Systems, Department of Electrical Engineering, Tsinghua University, Beijing 100084, China

**Keywords:** metal–organic frameworks (MOFs), organic linkers, ammonia, adsorption capacity, manure composting

## Abstract

Ammonia (NH_3_) is a common pollutant mostly derived from pig manure composting under humid conditions, and it is absolutely necessary to develop materials for ammonia removal with high stability and efficiency. To this end, metal–organic frameworks (MOFs) have received special attention because of their high selectivity of harmful gases in the air, resulting from their large surface area and high density of active sites, which can be tailored by appropriate modifications. Herein, two synthetic metal–organic frameworks (MOFs), 2-methylimidazole zinc salt (ZIF-8) and zinc-trimesic acid (ZnBTC), were selected for ammonia removal under humid conditions during composting. The two MOFs, with different organic linkers, exhibit fairly distinctive ammonia absorption behaviors under the same conditions. For the ZnBTC framework, the ammonia intake is 11.37 mmol/g at 298 K, nine times higher than that of the ZIF-8 framework (1.26 mmol/g). In combination with theoretical calculations, powder XRD patterns, FTIR, and BET surface area tests were conducted to reveal the absorption mechanisms of ammonia for the two materials. The adsorption of ammonia on the ZnBTC framework can be attributed to both physical and chemical adsorption. A strong coordination interaction exists between the nitrogen atom from the ammonia molecule and the zinc atom in the ZnBTC framework. In contrast, the absorption of ammonia in the ZIF-8 framework is mainly physical. The weak interaction between the ammonia molecule and the ZIF-8 framework mainly results from the inherent severely steric hindrance, which is related to the coordination mode of the imidazole ligands and the zinc atom of this framework. Therefore, this study provides a method for designing promising MOFs with appropriate organic linkers for the selective capture of ammonia during manure composting.

## 1. Introduction

Due to the worldwide intensive industrialized development of livestock and poultry breeding, a large amount of breeding wastes has been generated, posing a serious threat to the ecological environment and human and animal health. Thus, the reduction, detoxification, and resource utilization of livestock and poultry breeding waste have attracted much attention; composting technology has proven to be the most effective treatment method [1,2,3,4]. However, the applications of composting technology are limited due to the odor generated during the composting process. Constituted of complex gases, ammonia in particular, the odor not only harms the environment and health of humans and animals but also reduces the fertilizer efficiency and agricultural value of the compost [5,6,7]. Therefore, the effective control of and reduction in the emissions of odorous gases during the composting process play a vital role in achieving high-quality, efficient, and pollution-free composting given the urgent need for ecological and environmental protection.

In recent years, many practical methods, such as the adjustment of the compost preparation parameters [8,9], change in the aeration rate [10,11], and addition of external additives [12,13,14], have been developed to promote the composting process and reduce the production and discharge of odor during aerobic composting. Thus far, the external additives method has attracted widespread attention from scholars because of its good performance not only in promoting composting efficiency and reducing odor emissions but also in increasing organic matter conversion and reducing nitrogen losses [15,16]. Despite the fact that biochar is most commonly employed as an adsorbent among all external additives, this material has a low adsorption capacity and cannot be easily recovered, which prevents it from being the premium solution to the breeding waste processing problem.

Metal–organic frameworks (MOFs), on the other hand, given their composition and structure diversity, ultra-high specific surface area, high and adjustable porosity, and open metal sites, have received special attention regarding the selective capture of harmful gases from the air, especially ammonia, as reported by several studies [17,18,19,20,21,22]. As reported by Li et al. [23], the MOF material Cu(INA)_2_ has repeatable and remarkable ammonia adsorption and desorption capacities of up to 13 mmol/g due to its reversible structural transformation with the adsorption and desorption of ammonia. In addition, the new IL@MOF composite [BOHmim][Zn_2_Cl_5_]@MIL-101(Cr) developed by Zhong et al. [24] exhibits a record NH_3_ uptake of 24.12 mmol/g, which can be attributed to the multiple adsorption sites and large free volume for NH_3_ provided by the IL confined in the framework of the MOF. Moreover, an MOF-74 analog, M-MOF-74 (M = Zn, Co), was prepared and studied by Glover et al. [25], showing a strong ammonia removal ability of 7.60 mol kg^−1^ and 6.70 mol kg^−1^ for Mg-MOF-74 and Co-MOF-74, respectively, in dry conditions of 0% relative humidity (RH).

Regardless of the potential of MOF materials to reduce ammonia emissions, as exemplified above, their application in composting is rather limited due to the following: As part of the agricultural field, the additives and materials employed in composting should not be detrimental to the environment and soil and, therefore, should not introduce any harmful elements to the composting system, which screens out many MOF materials with heavy metal elements [26,27]. Moreover, the complex composting process is usually carried out in a humid and complex atmosphere, which requires the stability of MOF materials in such a harsh environment. Taking the multiple requirements of composting additives into consideration, a reasonable preparation strategy for MOF materials is urgently needed and the factors affecting NH_3_ removal must be analyzed in order to make full use of the structural advantages of MOFs and further improve their adsorption performance.

In this study, two MOFs, ZIF-8 and ZnBTC, with different organic linkers were designed and synthesized by a solvothermal method. Ammonia adsorption over ZIF-8 and ZnBTC was systematically measured, revealing the distinct ammonia adsorption capacities of ZIF-8 (1.26 mmol/g) and ZnBTC (11.37 mmol/g). Furthermore, surface area, porosity measurements, and DFT calculations were carried out to illuminate the different ammonia adsorption mechanisms between the two MOFs, confirming the significance of the judicious selection for organic linkers associated with different coordination modes and different exposed active sites.

## 2. Experimental Methods

### 2.1. Materials and Reagents

Zn(NO_3_)_2_·6H_2_O (AR, 99%) and ZnAc_2_·2H_2_O were obtained from Macklin Co., Ltd. (Shanghai, China). Dimethylglyoxaline dimethylimidazole and benzene-1,3,5-tricarboxylic acid were purchased from Sinopharm Chemical Reagent Co.,Ltd. (Shanghai, China). Sodium hydroxide (NaOH, 95%) was obtained from Beijing Huarong Chemical factory (Beijing, China). The DMF and H_2_O were commercially available and used as supplied without further purification.

### 2.2. Synthesis of ZIF-8 and ZnBTC

ZnBTC was prepared according to the methods reported in the literature [28,29,30]. The detailed preparation process is described as follows: First, 1.8 g of ZnAc_2_·6H_2_O was dissolved in 60 mL of deionized water with constant stirring for 20 min and named solution A. Then, 0.84 g of C_9_H_6_O_6_ and 0.48 g of NaOH were added to the mixture of the water and ethanol (3:1), stirred until evenly dissolved and named solution B. Solution B was slowly added to solution A under stirring conditions. Then, the mixture was transferred onto glass plates and dried in an oven at 120 °C for 12 h. After that, the autoclave was cooled down to room temperature. The resulting white crystal of Zn-BTC precipitate was filtered and washed with ethanol and deionized (DI) water several times until the filtrate presented as neutral. Finally, the Zn-BTC was dried in a vacuum oven at 60 °C for 3 h.

The ZIF-8 was prepared according to the methods reported in the literature [31].

### 2.3. Adsorption Experiments

The pure NH_3_ adsorption and desorption curves of the samples were determined by a static capacity sorption analyzer (Bei Shi De, BSD-PSPM, Beijing, China). Before testing, the samples were activated at 50 °C for at least 2 h until the mass no longer changed. For the adsorption experiments in the NH_3_/H_2_O system, about 20 mL of NH_3_/H_2_O (4:1, *v:v*) and 0.10 g of adsorbent were placed in a covered container. The adsorption during the composting process was implemented in a similar manner. The adsorbent was added at mesophilic and earlier thermophilic phases during the aerobic composting of manure. After the adsorption reached saturation, the mixture was added to a saturated solution of potassium chloride using a temperature-controlled shaker at room temperature. The mixed- solution was then filtered and analyzed using an Automatic Discrete Analyzer (SmartChem 140, Catania, Italy) to estimate the adsorption intake of NH_3_.

### 2.4. Characterization

The morphologies of the prepared MOFs were studied by field-emission scanning (SEM, S-4700, ISS Group Services Ltd., Manchester, England). Fourier transform infrared (FT-IR) spectroscopy was carried out by a Nicolet 6700FT-IR spectrophotometer (Thermo Fisher Scientific Inc., Waltham, MA, USA). The crystal structures of the prepared MOFs were fared, performed by an X-ray diffractometer (Bruker AXS D8-Advance, Bruker, Billerica, MA, USA) in the 2θ range from 5° to 40° at a scan rate of 10° min^−1^. N_2_ adsorption/desorption isotherms were recorded by a JWGB Sci & Tech Ltd (Beijing, China). The Brunauer–Emmett–Teller (BET) surface area of the prepared MOFs was tested through a volumetric method. The pure NH_3_ adsorption and desorption curves of the samples were determined by a static capacity sorption analyzer (BSD-PSPM, Microtrac, Duesseldorf, Germany), and the samples were activated at 50 °C for at least 2 h before testing or until the mass no longer changed. Theoretical calculations were carried out using the Quickstep algorithm of the CP2K package.26 to elucidate the adsorption behavior and inherent mechanism. The Perdew–Burke–Ernzerhof (PBE) function [32] with Grimme’s dispersion correction and with Becke–Johnson damping (D3BJ) [33,34] was employed for all the calculations. The Gaussian and plane wave methods were used, and the wave function was expanded in the Gaussian double-ζ valence polarized (DZVP) basis set. A convergence criterion of 3.0 × 10^−^^6^ a.u. was used for the optimization of the wave function. The adsorption energy of ammonia Δ*E*_n_ was calculated as follows:Δ*E*_n_ = *E*_ZIF8(n)_ − *E*_ZIF8(n−1)_ − *E*_NH__3_ (for ZIF8 system)(1)
Δ*E*_n_ = *E*_ZnBTC(n)_ − *E*_ZnBTC(n−1)_ − *E*_NH__3_ + *E*_H__2O_ (for ZnBTC system)(2)
where the *E*s with different subscripts are the energies of each species obtained from DFT calculations, and n refers to the nth NH_3_ molecule that was adsorbed onto the MOF.

## 3. Results and Discussion

### 3.1. Characterization of MOF Materials

As shown in Figure 1, two types of Zn-based MOFs with different organic ligands, ZnBTC and ZIF-8, were synthesized by a solvothermal method. The ZnAc_2_·6H_2_O and H_3_BTC were dissolved in the selected solvents, and the resulting solutions were mixed and then placed in the Teflon-lined stainless steel autoclave. The stirring lasted for 9 h at 150 °C. Grayish-white crystals were obtained by washing with deionized water and drying at ambient temperature.

In the reaction systems, infinite Zn chains are interconnected by the organic ligand linkers into a three-dimensional microporous framework. Due to their porous structures with different functionalized channels, the ZnBTC and ZIF-8 show promise for use as adsorption materials.

The morphologies and microstructures of the as-prepared two MOFs were characterized by scanning electron microscopy (SEM). As shown in Figure 2, both ZIF-8 and ZnBTC possess smooth surfaces. However, the two MOFs exhibit quite different morphologies and structures, which can be attributed to the different coordination modes of the different organic ligands. A relatively homogenous and hexagonal-structured morphology was observed for ZIF-8, while ZnBTC has nonuniform rod-like structures with lengths ranging from 0.5 to 3 μm. As exemplified by the calculations, one zinc ion is coordinated by four imidazole ligands in ZIF-8 and adopts a tetragon coordination mode. ZnBTC employs an octagon coordination mode, which can be easily attacked by polar molecules.

The powder XRD patterns for both ZIF-8 and ZnBTC were obtained before and after the capture of NH_3_. As shown in Figure 3, the PXRD patterns of the pristine ZIF-8 and ZnBTC are well-matched with those reported [30,32], revealing the crystalline nature of ZIF-8 and ZnBTC. In detail, the characteristic peaks of ZIF-8 at 2θ = 7.3°, 10.3°, 12.7, 14.8, 16.4, 18.0, 24.6, and 26.7 can be attributed to the crystalline planes (011), (002), (112), (022), (013), (222), (233), and (134), respectively. The as-prepared ZnBTC displayed an intense diffraction peak at 2θ = 10°, consistent with that in the literature [35], confirming the formation of ZnBTC. Nevertheless, the PXRD of ZnBTC significantly changed after the co-adsorption of H_2_O/NH_3_, and it was speculated that NH_3_ and H_2_O destroyed the structure of ZnBTC to some extent. From the PXRD of ZIF-8 before and after adsorption of NH_3_, it can be seen that the diffraction peak of ZIF-8 does not obviously change, indicating that NH_3_ has less of an effect on ZIF-8.

In order to gain insights into the ammonia adsorption mechanisms of the two MOF materials, the FT-IR spectra were measured before and after NH_3_ absorption. As shown in Figure 3b, compared with that of pristine ZnBTC material, two additional peaks at 3300 and 1276 cm^−1^, which correspond to ν (N–H) and δ (Zn-NH_3_) [36], respectively, can be observed after the absorption of NH_3_. In addition, the large, broad peak in the range of 3200–3600 cm^−1^ was attributed to the absorbed H_2_O. The observed results suggested obvious interactions between the zinc ions in ZnBTC and the absorbed ammonia molecules. No apparent change in the absorption signals was observed for ZIF-8 after the ammonia absorption, indicating relatively weak interactions between the zinc ions in ZIF-8 and the ammonia molecules. The stark contrast of the two MOF materials before and after ammonia absorption revealed the effect of different ligands on the interactions between the absorbed ammonia molecules and the MOF materials. For ZIF-8, the absorption peaks of 3134 and 2927 cm^−1^ are C–H stretching vibration peaks of aromatic and aliphatic groups in the imidazole ligand, and the peak at 1573 cm^−1^ is due to the stretching vibration of C=N on the imidazole ring. The signals within the range from 1500 to 1350 cm^−1^ and those from 1500 to 600 cm^−1^ could be attributed to the stretching vibration and bending vibration of the imidazole ring, respectively. The peak at 422 cm^−1^ is the stretching vibration peak of Zn–N [37].

### 3.2. NH_3_ Adsorption Studies

The NH_3_ adsorption and desorption curves of ZnBTC and ZIF-8 were investigated to evaluate the adsorption capacity. As shown in Figure 4, ZnBTC exhibited an adsorption capacity of 113.77 mL/g (5.04 mmol/g) at low pressure. The excellent adsorption capacity of the ZnBTC prepared in this study may be due to the metal vacancy coordination created in the ZnBTC structure, which facilitated the NH_3_ to enter the pores and be quickly adsorbed in the vacant coordination. With the increase in pressure, NH_3_ was adsorbed in the void, and the final adsorption capacity was as high as 254.88 mL/g (11.37 mmol/g). The NH_3_ desorption curve of ZnBTC shows that some of the NH_3_ was desorbed with the decrease in pressure, and this part of NH_3_ should be absorbed with physical effects. However, most of the NH_3_ was still not desorbed, which means that this part of the NH_3_ gas was chemisorbed and absorbed inside the structure. Compared with ZnBTC, the adsorption capacity of ZIF-8 is low, and all the NH_3_ would have been removed from the pore structure with the decrease in pressure. This may have occurred mainly because there are no metal vacancies and adsorption functional groups inside the structure of MOFs.

To further demonstrate the application of the MOFs for the absorption of NH_3_, adsorption experiments were also carried out under ammonia water during the composting process. The adsorption performance during the thermophilic phases during aerobic composting was defined as ZIF-8-1 and ZnBTC-1. ZIF-8-2 and ZnBTC-2 represent the adsorption performance under a 25% NH_3_ solution steam atmosphere. ZIF-8-3 and ZnBTC-3 were tested with pure ammonia for comparison. It can be seen that the adsorption capacity of ZnBTC is better than that of ZIF-8, as shown in Figure 5, which indicates that the adsorption performance of ZnBTC maintains superiority in contrast with ZIF-8 under the same conditions. In addition, it can be observed that the adsorption capacity during the composting process was lower than that in ammonia water and pure ammonia. The performance parameters of adsorption capacity under pure ammonia were the highest, followed by the 25% NH_3_ solution steam atmosphere during aerobic composting. As can be seen, ZnBTC-3 and ZnBTC-2 show excellent adsorption capacities of 11.37 and 6.04 mmol/g, respectively. The adsorption capacities of ZIF-8-3 (1.26 mmol/g) and ZIF-8-2 (0.09 mmol/g) were much lower. The adsorption capacities of ZnBTC-1 (2.55) and ZIF-8-1 (undetected) during aerobic composting were the lowest. There may have been two reasons for these findings: On the one hand, the adsorption capacity of ammonia increases with its concentration. On the other hand, competitive adsorption may exist between NH_3_ and other gases, such as H_2_O under a 25% NH_3_ solution steam atmosphere, and the H_2_S and N_2_O in the composting process occupied the adsorption sites of MOF materials.

### 3.3. Adsorption Mechanism

To further characterize the porosity of the two MOFs, the N_2_ adsorption–desorption isotherms of samples were measured by the static method at 77 K. The specific surface area and pore structure of ZnBTC and ZIF-8 materials were measured by the automatic rapid specific surface and pore analyzer (Microtrac, Duesseldorf, Germany) (as shown in Figure 6). The built-in software of the instrument was used. The specific surface area of the material was calculated by the BET or Langmuir model. Before the test, about 90 mg of the MOF material was weighed and heated for more than 150 h under vacuum conditions to remove the moisture and impurities in the material.

As a result, we found the ZnBTC and ZIF-8 possess a Brunauer–Emmett–Teller (BET) surface area of 1298.50 and 44.47 m^2^ g^−1^, respectively. The surface area of ZIF-8 is apparently higher than that of ZnBTC. Hence, research is needed to determine whether it is the specific surface area or adsorption strength that determines the adsorption capacity of ammonia gas.

The density functional theory calculation was then performed to elucidate the adsorption behavior of the two materials and their inherent mechanism. Rather than simply calculating the adsorption of a single ammonia molecule, we made the two MOF materials gradually adsorb one to four ammonia molecules and calculated the adsorption energy of each adsorbed ammonia molecule. Initially, ZIF-8 was directly employed in the reported crystal structure (CCDC No. 602542), while ZnBTC was modified from the structure (CCDC No. 963916) in which the nitrate ligand was exchanged for chloride for easy computation, and the H_2_O ligand was maintained. The calculated results showed that each NH_3_ molecule adsorbed onto ZnBTC replaced the H_2_O ligand and led to a negative binding energy. Therefore, the NH_3_ gas adsorbed by ZnBTC is stable (Figure 7b). Comparably, the first two NH_3_ molecules seemed quite difficult to adsorb onto ZIF-8 due to extremely high binding energy (Figure 7a). The metal–ligand binding could also be revealed by the bond length (Figure 7c). Remarkably, the bond length in ZIF-8 is more than 4 Å, which is far beyond any coordination bond length. In contrast, the bond length in ZnBTC is stably localized at 2.08 Å, a typical coordination bond length. Subsequently, the average binding energy of ZIF-8 is larger than zero, indicating that ZIF-8 is weak for adsorbing NH_3_. The adsorbability difference between ZIF-8 and ZnBTC results from their crystal structures. The zinc in ZnBTC is hexacoordinated, and the coordination of NH_3_ instead of H_2_O was sterically-allowed. In contrast, the zinc in ZIF-8 is tetracoordinated, and there is less room for an extra NH_3_. As a total consequence, ZnBTC exhibited supreme NH_3_ adsorbability (Figure 7d).

## 4. Conclusions

In summary, two kinds of MOFs with different organic ligands were designed and analyzed with experimental and theoretical calculations. XRD, FTIR, and other tests proved that the MOFs were successfully prepared. The ZnBTC showed a higher NH_3_ uptake of 11.37 mmol/g than ZIF-8 (1.26 mmol/g) due to the multiple adsorption sites, which helps the composting process reduce environmental pollution. The theoretical calculation also showed that ZnBTC has a higher adsorption capacity than ZIF-8. Such experimental adsorption capacities of MOFs were sufficiently consistent with the theoretical calculation. As far as we are aware, we are the first to design structural systems suitable for the materials that are available for the capturing of NH_3_ from the composting process. These results not only provide a candidate method for NH_3_ abatement but also provide a theoretical basis for material design for specific adsorption targets in the composting process.

## Figures and Tables

**Figure 1 molecules-27-05615-f001:**
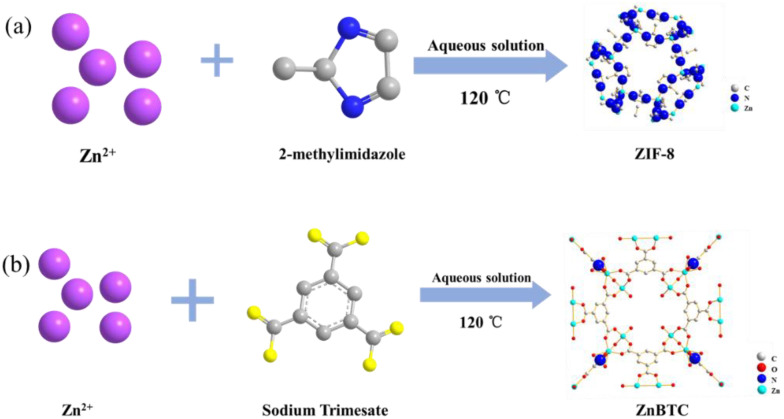
Diagrammatic flow charts of ZIF-8 (**a**) and ZnBTC (**b**).

**Figure 2 molecules-27-05615-f002:**
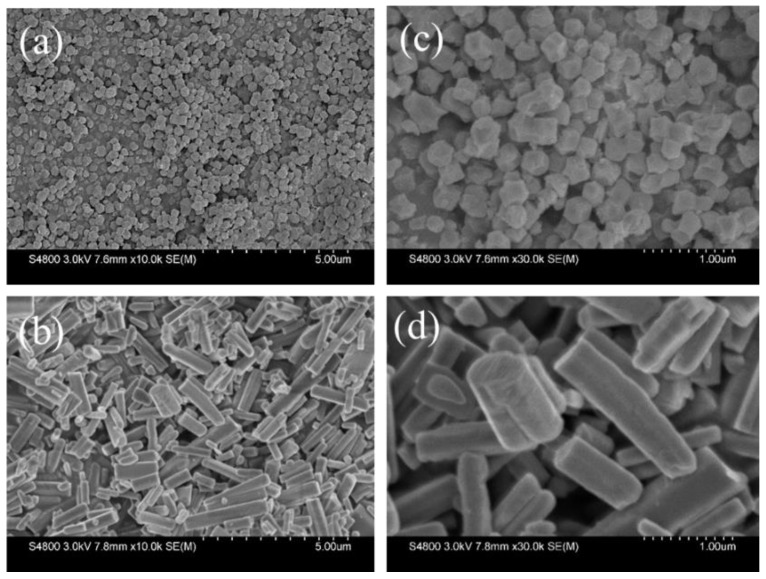
SEM images of the ZIF-8 (**a**) and ZnBTC (**b**); enlarged images of ZIF-8 (**c**) and ZnBTC (**d**).

**Figure 3 molecules-27-05615-f003:**
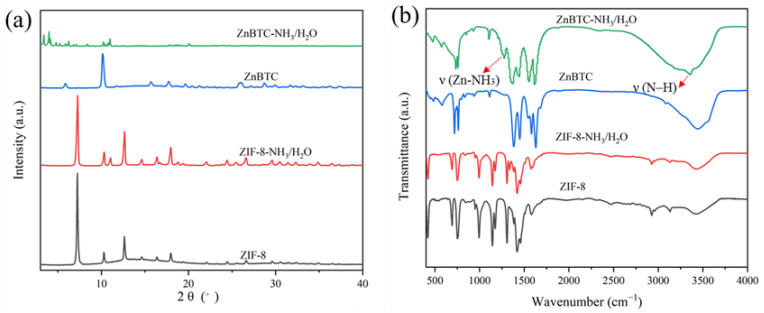
The XRD patterns (**a**) and FTIR (**b**) of ZIF-8 and ZnBTC.

**Figure 4 molecules-27-05615-f004:**
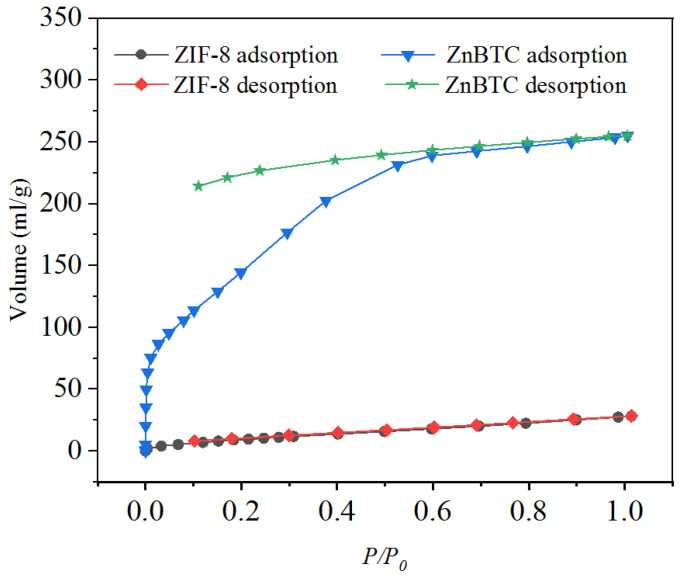
NH_3_ adsorption and desorption curves obtained for ZnBTC and ZIF-8.

**Figure 5 molecules-27-05615-f005:**
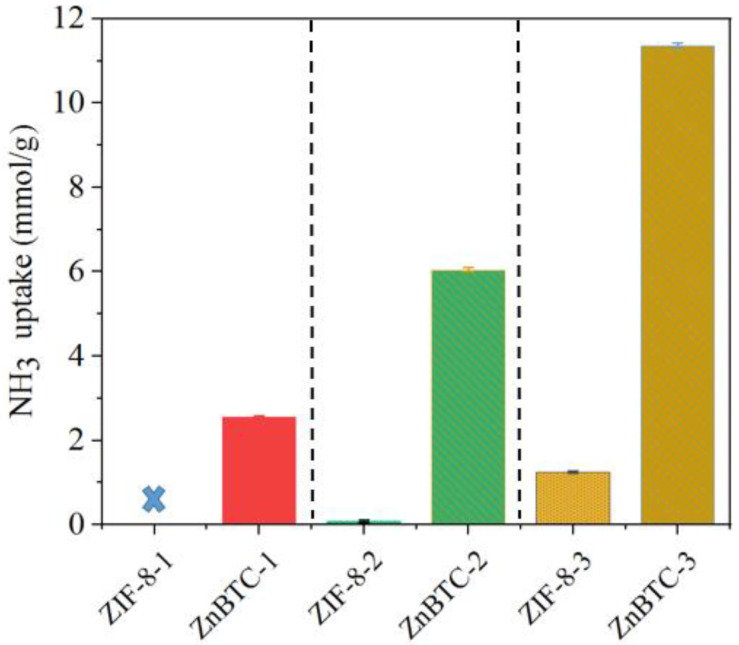
NH_3_ adsorption capacity of ZIF-8 and ZnBTC in different settings.

**Figure 6 molecules-27-05615-f006:**
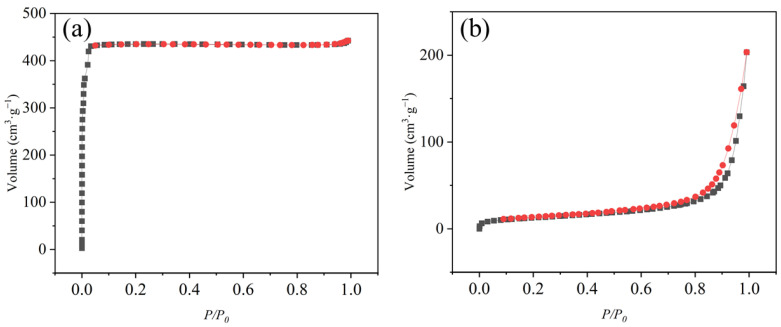
N_2_ adsorption−desorption isotherms of (**a**) ZIF-8 (**b**) ZnBTC.

**Figure 7 molecules-27-05615-f007:**
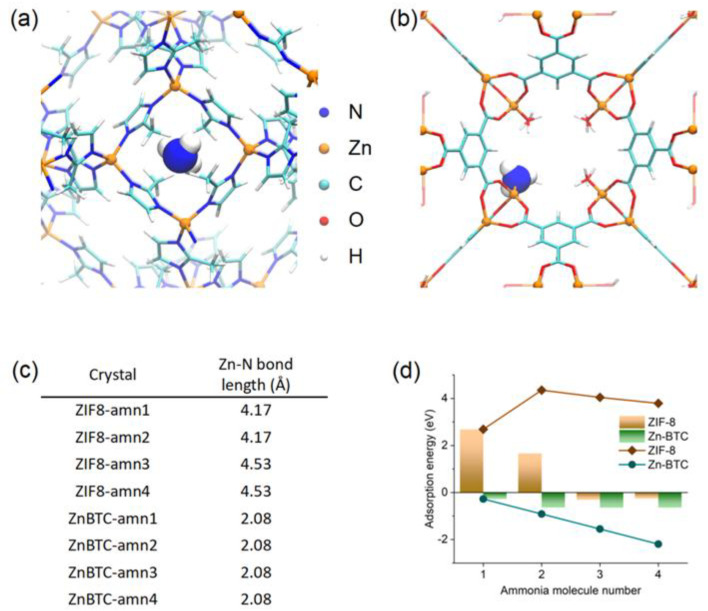
DFT calculation results of ammonia adsorption. (**a**) The structure of ZIF-8 with one adsorbed ammonia molecule (ZIF-8-amn1); (**b**) the structure of ZnBTC with one adsorbed ammonia molecule (ZnBTC-amn1), both of which were drawn by VMD (1.9.3) package [34]; (**c**) the Zn-N bond length in each adsorption structure; (**d**) the adsorption energy (column) and cumulative adsorption energy (line) of each crystal material.

## Data Availability

The authors do not have permission to share data.

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
