# Peer review of "A Combined Experimental and Computational Study on the Adsorption Sites of Zinc-Based MOFs for Efficient Ammonia Capture"

_molecules, 2022, doi:10.3390/molecules27175615_

Round 1

Reviewer 1 Report

Author carried out combined experimental and computational investigations on the adsorption sites of Zinc-based MOFs to capture ammonia. There are various ambiguities stated below due to which I can’t recommend to publish it in present form.

Major Comments

1. Line 265-267: It is reported in theoretical study that “ZIF-8 was directly employed in the reported crystal structure (CCDC No. 265 602542), while ZnBTC was modified from the structure (CCDC No. 963916)”. This is not clear that either structures of these Zn based MOFs with organic linkers are reported before by authors?

2. Authors should describe clearly about the novelty of work if MOFs structures are reported before with proper citations.

3. Line 184-186: In description author stated “PXRD of ZnBTC changed significantly after the co-adsorption of H2O/NH3, and it was speculated that NH3 and H2O destroyed the structure of ZnBTC to some extent”. How it was confirmed that these change in PXRD pattern is solely due to co adsorption? There should be detailed account on how PXRD was carried out while co adsorption of H2O/NH3?

4. The explanation of FTIR data w.r.t adsorption of ammonia is not coherent and unclear.

5. How adsorption capacities were calculated?

6. Outcomes of computational investigations are not summarized. & how this study is accounting for experimental results is totally unclear.

7. Manuscript needs major overhaul to for grammar and structure of manuscript.

8. Write either Zn- BTC or ZnBTC

Author Response

  1. Line 265-267: It is reported in theoretical study that “ZIF-8 was directly employed in the reported crystal structure (CCDC No. 265 602542), while ZnBTC was modified from the structure (CCDC No. 963916)”. This is not clear that either structures of these Zn based MOFs with organic linkers are reported before by authors?

Response: Thank you for your comments. The structures of these two Zn based MOFs with organic linkers are reported before by other researcher. However, the adsorption properties have not been studied systematically, especially the adsorption properties of ammonia gas.

  1. uthors should describe clearly about the novelty of work if MOFs structures are reported before with proper citations.

Response: Thank you for your comments. The innovation of this work lies in the systematic analysis and comparison of the adsorption properties of two materials for the removal of ammonia gas. The reference of the reported MOFs structures have been added in this paper. Your comment is greatly appreciated to improve the quality of the paper. As suggested by your comment, large amounts of changes of articles have been made in the revised version. Innovativeness was highlighted in our revised manuscript.

  1. ine 184-186: In description author stated “PXRD of ZnBTC changed significantly after the co-adsorption of H2O/NH3, and it was speculated that NH3 and H2O destroyed the structure of ZnBTC to some extent”. How it was confirmed that these change in PXRD pattern is solely due to co adsorption? There should be detailed account on how PXRD was carried out while co adsorption of H2O/NH3?

Response: Thanks for your opinions. We have enriched the description of the detailed account on how PXRD was carried out, mainly from the experiment and test characterization process according to your suggestion. The revised manuscript was as follows:

From the PXRD of ZIF-8 before and after adsorption of NH3, it can be seen that the diffraction peak of ZIF-8 does not change obviously, indicating that NH3 has less effect on ZIF-8.

  1. The explanation of FTIR data w.r.t adsorption of ammonia is not coherent and unclear.

Response: Thanks for your opinions. We provide a more detailed explanation of FTIR data w.r.t adsorption of ammonia.

We have checked and revised them as follows:

In addition, the large broad peak in the range of 3200-3600 cm-1 is attributed to the absorbed H2O.

  1. How adsorption capacities were calculated?

Response: Thanks for your opinions. The adsorption capacities was achieved by evaluating the adsorption intake of NH3 of the mixed-solution, which was extracted of ammonia from the saturated adsorption sample in a solution of potassium chloride. The detailed steps are as follows. Activated sample was put into a NH3/H2O (4:1, v:v) steam atmosphere. The weight of the sample was measured until it did not change evidently. The samples were then washed with water, and the content of NH3 can be obtained by Automatic Discrete Analyzer (SmartChem 140).

  1. utcomes of computational investigations are not summarized. & how this study is accounting for experimental results is totally unclear.

Response: Thank you for your suggestions, we have supplemented the summary of the calculations and the experimental results are explained in detail. The revised manuscript was as follows:

The theoretical calculation also manifests that ZnBTC owns higher adsorption capacity than ZIF-8.

  1. Manuscript needs major overhaul to for grammar and structure of manuscript.

Response: Thank you for your suggestions, some minor grammatical issues were corrected and marked red in our revised manuscript.

  1. Write either Zn- BTC or ZnBTC

Response: Sorry for the mistake, we have read the paper carefully and revised them.

Reviewer 2 Report

This article has done a combined experimental and theoretical investigation of two MOFs: ZIF-8 and ZNBTC to evaluate their NH3 adsorption performance for efficient NH3 capture. The synthesis, BET adsorption characterization, FT-IR, XRD, and DFT calculation have provided abundant information on these two materials. There are some questions regarding the DFT calculation section.

1. Line 138. Did the author include DJ damping or not? If BJ damping is not included, it should be written as “Grimme’s dispersion correction (D3)”. If BJ damping is included, it should be written as “Grimme’s dispersion correction with Becke-Johnson damping (D3BJ)”.

2. The author should cite the PBE exchange-correlation functional (Phys. Rev. Lett., 77 (1996) 3865-68.), and the D3 correction (J. Chem. Phys., 132 (2010) 154104.) or D3BJ (J. Comp. Chem. 32 (2011) 1456-65.).

3. Are structures in Figure 7 visualized by VMD? The author should properly cite the visualization software.

4. Line 260. Please check and revise the expression here “The density functional theory calculation was then performed by density functional theory calculation”.

5. For Figure 1 and 7, it would be better if the author can mention what kind of atom the color ball represents in the caption.

6. ZnBTC was modified from ZnBTC single crystal data in which the nitrate ligand was replaced by Cl. I doubt the validity of the modification. It will not require too much additional computational power using [NO3]- ligand, even if the O on NO3 is virtual occupation (position not determined). DFT can still optimize it.

7. How is the XRD of DFT optimized ZnBTC compared to the experimental XRD?

8. I understand that Eads on ZIF-8 is positive since all terminations are saturated, such -CH3 and -H. There is no spatial attachment between N and Zn in ZIF-8. In ZnTBC, there is also no direct attachment between N and Zn if H2O is not replaced by NH3. The author has proposed that NH3 is replacing the H2O ligand in ZnBTC, as reflected in Figure 7b and Line 144. Does the comparison of Eads between ZIF-8 and ZnTBC fair enough? As for the replacement pathway, there might be a very large activation energy barrier. The transition state may involve the coexistence of H2O and NH3 near a single Zn. Is the replacement pathway consistent with the experimental data?

Author Response

  1. Line 138. Did the author include DJ damping or not? If BJ damping is not included, it should be written as “Grimme’s dispersion correction (D3)”. If BJ damping is included, it should be written as “Grimme’s dispersion correction with Becke-Johnson damping (D3BJ)”.

Response: Many thanks for this reminding by this reviewer. We included the DJ damping in the calculation. The description in the manuscript is revised accordingly.

  1. The author should cite the PBE exchange-correlation functional (Phys. Rev. Lett., 77 (1996) 3865-68.), and the D3 correction (J. Chem. Phys., 132 (2010) 154104.) or D3BJ (J. Comp. Chem. 32 (2011) 1456-65.).

Response: We are highly appreciated that the reviewer provided the necessary citations which greatly improves the quality of this work. These citations are appended in the revised manuscript.  

  1. Are structures in Figure 7 visualized by VMD? The author should properly cite the visualization software.

Response: The corresponding citation (J. Molec. Graphics 1996, 14.1, 33-38) is appended in the revised work.

  1. Line 260. Please check and revise the expression here “The density functional theory calculation was then performed by density functional theory calculation”.

Response: The sentence is corrected in the revised work.

  1. For Figure 1 and 7, it would be better if the author can mention what kind of atom the color ball represents in the caption.

Response: A lot of thanks for this suggestion from this reviewer. We appended the color instructions in Figure 7.

  1. ZnBTC was modified from ZnBTC single crystal data in which the nitrate ligand was replaced by Cl. I doubt the validity of the modification. It will not require too much additional computational power using [NO3]-ligand, even if the O on NO3 is virtual occupation (position not determined). DFT can still optimize it.

Response: We are thankful for this valuable comment by this reviewer. Actually, the calculation in this work aims to explain why ZnBTC are advantageous in adsorbing NH3 from the view of energy variation. Since the ligand (NO3 or Cl) and ammonia are anchored on different zinc ions, the ligand exchange has trivial effect on the ammonia adsorption, and the ligand modification is thus valid for the calculation. On the other hand, The DFT computational power are not only affected by the atomic number, but the electronic structures. The twelve [NO3] ligands in the cell have more complicated electronic structure than Cl ligands, which seems to makes the SCF cycle difficult to converge.

  1. How is the XRD of DFT optimized ZnBTC compared to the experimental XRD?

Response: We supported both simulated XRD patterns of ZIF8, ZIF8-NH3, ZnBTC, and ZnBTC-NH3., in which ZIF8 consists well with experiments and ZnBTC also shows character signals. However, the difference between the crystal with and without NH3 adsorption is not significant. This is probably because the amount of NH3 in calculation is much less than experiment.

  1. I understand thatEads on ZIF-8 is positive since all terminations are saturated, such -CH3 and -H. There is no spatial attachment between N and Zn in ZIF-8. In ZnTBC, there is also no direct attachment between N and Zn if H2O is not replaced by NH3. The author has proposed that NH3 is replacing the H2O ligand in ZnBTC, as reflected in Figure 7b and Line 144. Does the comparison of Eads between ZIF-8 and ZnTBC fair enough? As for the replacement pathway, there might be a very large activation energy barrier. The transition state may involve the coexistence of H2O and NH3 near a single Zn. Is the replacement pathway consistent with the experimental data?

Response: We agree with this reviewer’s comment that there might be energy barrier on the molecule replacement theoretically. However, it is widely known that zinc hydrate ions may spontaneously transfer to zinc ammonia complex in water solution and room temperature provided a certain concentration of ammonia (see below reaction [1] and [2] ). Therefore, it may be deduced that the energy barrier on the replacement reaction from zinc water complex to zinc ammonia complex is small enough that the ammonia adsorption could be spontaneously occurred.

Zn(H2O)62+ + 2NH3= Zn(OH)2 + 2NH4+ + 2H2O                                               [1]

Zn(OH)2 + 4NH3 = [Zn(NH3)4]2+ + 2OH                                                          [2]

Round 2

Reviewer 1 Report

no more comments, authors revised appropriately.

Reviewer 2 Report

The substitution of H2O with NH3 is interesting, and I accept the authors' explanation. I am looking forward to a more in-depth future study of this system from the author.

At this stage, the paper can be accepted in its present form.